# Mechanical Properties of Ultra-High Performance Concrete before and after Exposure to High Temperatures

**DOI:** 10.3390/ma13030770

**Published:** 2020-02-07

**Authors:** How-Ji Chen, Yi-Lin Yu, Chao-Wei Tang

**Affiliations:** 1Department of Civil Engineering, National Chung-Hsing University, No. 250, Kuo Kuang Road, Taichung 402, Taiwan; hojichen@dragon.nchu.edu.tw (H.-J.C.); n0937699278@gmail.com (Y.-L.Y.); 2Department of Civil Engineering & Geomatics, Cheng Shiu University, No. 840, Chengching Rd., Niaosong District, Kaohsiung 83347, Taiwan; 3Center for Environmental Toxin and Emerging-Contaminant Research, Cheng Shiu University, No. 840, Chengching Rd., Niaosong District, Kaohsiung 83347, Taiwan; 4Super Micro Mass Research & Technology Center, Cheng Shiu University, No. 840, Chengching Rd., Niaosong District, Kaohsiung 83347, Taiwan

**Keywords:** ultra-high performance concrete, residual mechanical properties, high temperature

## Abstract

Compared with ordinary concrete, ultra-high performance concrete (UHPC) has excellent toughness and better impact resistance. Under high temperatures, the microstructure and mechanical properties of UHPC may seriously deteriorate. As such, we first explored the properties of UHPC with a designed 28-day compressive strength of 120 MPa or higher in the fresh mix phase, and measured its hardened mechanical properties at seven days. The test variables included: the type of cementing material and the mixing ratio (silica ash, ultra-fine silicon powder), the type of fiber (steel fiber, polypropylene fiber), and the fiber content (volume percentage). In addition to the UHPC of the experimental group, pure concrete was used as the control group in the experiment; no fiber or supplementary cementitious materials (silica ash, ultra-fine silicon powder) were added to enable comparison and discussion and analysis. Then, the UHPC-1 specimens of the experimental group were selected for further compressive, flexural, and splitting strength tests and SEM observations after exposure to different target temperatures in an electric furnace. The test results show that at room temperature, the 56-day compressive strength of the UHPC-1 mix was 155.8 MPa, which is higher than the >150 MPa general compressive strength requirement for ultra-high-performance concrete. The residual compressive strength, flexural strength, and splitting strength of the UHPC-1 specimen after exposure to 300, 400, and 500 °C did not decrease significantly, and even increased due to the drying effect of heating. However, when the temperature was 600 °C, spalling occurred, so the residual mechanical strength rapidly declined. SEM observations confirmed that polypropylene fibers melted at high temperatures, thereby forming other channels that helped to reduce the internal vapor pressure of the UHPC and maintain a certain residual strength.

## 1. Introduction

Ultra-high performance concrete (UHPC) is a cementitious composite characterized by high compressive strength, typically greater than 120 MPa, specific durability, tensile ductility, and toughness [1,2,3,4,5]. In the basic composition of UHPC, the aggregate is a filler and the binder reacts with water to form a matrix with cementing ability and a lubricating effect. In addition to lubricating the aggregate particles and providing proper workability, the matrix must fully solidify the aggregate particles to ensure the engineering properties of the hardened concrete. Considering the high strength and durability requirements of UHPC, the packing density of the particles must be increased. To ensure the sufficient strength of UHPC, silica fume and ultra-fine silica powder are often used in the matrix composition to increase the packing density of the particle system, thereby increasing the strength of the matrix. In addition to supplementary cementitious materials, a thickening agent is added, as is a suitable amount of a superplasticizer to ensure the proper fluidity and viscosity of the prepared matrix. Fibers are blended to increase the tensile and spalling resistance of the matrix. Because UHPC has many advantages, its application has been increasing since the advent of its research and development. For example, UHPC is now widely used in buildings, roads, bridges, and other structures. Among them, the application in bridges is the most widespread [6].

Basically, the overall cost of a structure is directly related to the cross-sectional dimensions of the structural members. The use of UHPC structural members reduces the cross-sectional dimensions and provides additional useful space in the building [7]. UHPC blended with fibers has excellent crack resistance and can suppress crack growth to reduce maintenance costs [8,9]. Although the cement content required in UHPC is higher, the material cost is still lower than that of ordinary concrete members requiring a larger cross-section. In addition, coarse aggregates are not used in UHPC and the amount of fine aggregate can be reduced to 30% [10]. UHPC can reduce material costs by approximately 56% compared with ordinary concrete, and energy consumption can be significantly reduced [11]. UHPC’s high-strength properties allow for the design of thinner structures that reduce the weight of the structure (less material used). This also reduces the amount of waste that will need to be dismantled in the future, thereby reducing transportation requirements and mitigating environmental impacts [7]. The use of by-products (such as fly ash and silica fume) instead of cement makes UHPC more sustainable [12]. Due to their excellent durability, UHPC components require less maintenance costs, so their life cycle costs can be reduced while providing longer service life [11]. Similarly, neighboring communities are not subject to routine maintenance or replacement facility operations, having positive social effects. In UHPC, not all cement has been hydrated. Therefore, the recycling of UHPC can be more effective because unhydrated cement can be used for further reactions [12]. Notably, UHPC is less harmful to the environment, having less of an impact on the ozone layer and producing fewer greenhouse gas emissions [13,14,15,16,17,18]. From this point of view, UHPC has a better impact on the economy, society, and the environment, and its application can lead to more sustainable construction.

Under the action of high temperature, the microstructure and properties of cement hydration products change, which directly or indirectly changes the macroscopic properties of the substrate, and affects the overall behavior of hardened concrete [19,20]. Due to the low porosity inside the UHPC, the release of vapor pressure under fire and high temperature can be easily hindered. Therefore, UHPC structures may be more susceptible to fire and high temperatures, resulting in tangible damage [21]. However, the use of polypropylene (PP) fibers can avoid comminuted damage or spalling [22,23,24,25]. The literature shows that the addition of 0.6% PP (volume ratio) can improve the fire resistance of UHPC, preventing spalling [26,27]. This is attributed the PP, after melting at high temperatures, providing an escape passage for the steam to release the accumulated pressure. However, on the surface of the sample, 0.3 to 0.5 mm cracks were observed. In addition, due to the dehydration of calcium silicate hydrates (C–S–H) colloid and the chemical decomposition and thermal expansion damage of UHPC materials, the mechanical properties of UHPC degraded [21]. At a high temperature of 1000 °C, UHPC with 0.6% PP added demonstrated a weight loss of less than 9% [26]. The results of Tai et al. [28] showed that the compressive strength of UHPC continued to increase with increasing temperature to 300 °C; however, once the temperature exceeded 300 °C, the compressive strength showed a downward trend. At high temperatures, the mechanical degradation of UHPC occurs mainly due to the weakening of the internal microstructure [29]. However, the addition of steel fibers to UHPC improved their behavior at elevated temperatures [28,29].

As mentioned above, UHPC is a more a sustainable material due to its improved durability, ecological factors, economic benefits, and recycling capabilities for a variety of applications. As a result, the use of UHPC in infrastructure projects has increased in recent years. Fire damage is a special characteristic failure mode of concrete structures because it has a considerable impact on the overall safety of the structure. Basically, the failure of structural concrete in a fire varies depending on the nature of the fire, the loading system, and the type of structure. The behavior of concrete during a fire mainly depends on its mix proportions and constituents and is determined by complex physicochemical transformations occurring in heating [30]. When exposed to high temperatures, ordinary-strength concrete and high-performance concrete follow similar trends in microstructure, but UHPC behaves differently. When exposed to high temperatures, UHPC is prone to explosive spalling due to its increased packing density. Therefore, exploring the influence of high temperature and fire damage on UHPC is essential. As such, we aimed to investigate the mechanical properties of UHPC before and after exposure to high temperatures. We found effective methods to prevent UHPC from spalling while maintaining its fresh properties. SEM observations of the UHPC specimens after exposure to different target temperatures in an electric furnace were performed to understand changes in their microscopic properties.

## 2. Experimental Details

### 2.1. Materials

The materials used in the tests included cement, silica fume, ultra-fine silica powder, fine aggregate, superplasticizer, viscous agent, steel fiber, and polypropylene fiber. A locally produced Type I Portland cement with a specific gravity of 3.15 and a fineness of 3400 cm^2^/g was used. The silica fume was produced locally, with a specific gravity of 2.1 and a silicon dioxide content of 92.4%. The ultra-fine silica powder was purchased from Dawei Stone Industry Co. (Jhubei, Taiwan). It had a specific gravity of 2.73 and an average particle diameter of 0.075 to 0.225 μm. The fine aggregate contained two different sizes of quartz sand (Type I and Type II). The physical properties and chemical composition of the fine particles are shown in Table 1, and the particle size distribution is provided in Table 2. The mixing ratio of the fine aggregate was 80% of Type I and 20% of Type II. Superplasticizers and viscous agent were purchased from Dawei Stone Industry Co. (Jhubei, Taiwan) (in accordance with the Chinese National Standards or the American Society for Testing Materials). The fibers used included steel and polypropylene fibers, as shown in Figure 1. The basic properties of these two fibers are listed in Table 3.

### 2.2. Mix Proportions and Casting of Specimens

Table 4 describes the mix design of the concrete used, including the control group and the experimental group. The control group (Mix No. HPC-1) had no fiber or special cementitious materials (silica fume and ultra-fine silica powder) for comparison and discussion. The experimental group had five different UHPC mix proportions. In the experimental group, the four important parameters were: the percentage of cement replaced by silica fume (SF), the percentage of cement replaced by ultra-fine silica powder (SFP), the amount of steel fiber (volume percent), and the amount of polypropylene fiber (volume percentage). Table 5 lists the percentage of these parameters used in the mix proportions of the UHPC. The amounts of superplasticizers and viscous agent were 2.6% and 0.1% by weight of the cementitious materials, respectively. The ratio of the weight of the fine aggregate to the weight of the cementitious materials was maintained at 1.28 in both the control and the experimental groups.

The concrete test proportions of each group were mixed using a forced single-shaft mixer, and the mixing procedure was as follows:(1)Before mixing, the fine aggregates were treated in a dry state because their water absorption rate approached zero.(2)We placed the cementitious materials, aggregates, and fibers (steel fiber and polypropylene fiber) into a mixing tank, and then dry blended uniformly for 5 min.(3)We poured the water into the mixer. After mixing for about 1 min, we poured all the superplasticizer and viscous agent into the mixture and mixed for another 4 min to produce fresh concrete.

After the concrete mixture was uniformly mixed, the fresh properties of each mixture were measured and recorded. Afterward, concrete specimens required for each test were cast and compacted using an external vibrator to ensure that the specimens were sufficiently compacted. The compressive strength and elastic modulus tests were conducted with cylindrical specimens with a diameter of 100 mm and a height of 200 mm. For the splitting strength test, we used a cylindrical specimen with a diameter of 150 mm and a height of 300 mm. For the flexural strength test, a prism sample 360 mm long, 100 mm wide, and 100 mm thick was used. After casting, all the specimens were covered with polyethylene sheets for 24 h overnight. The demolding operation was then conducted and each specimen was placed in a laboratory water bath until the day before the mechanical test.

### 2.3. Test Methods and Instrumentation

The UHPC tests of unit weight, slump, slump flow, compressive strength, elastic modulus, splitting tensile strength, and flexural strength were conducted according to the ASTM specifications listed in Table 6 [31,32,33,34,35,36,37]. To study the effect of high temperature on the mechanical properties of UHPC, the specimens were heated at the prescribed rate (2 °C/min) until the temperature in the furnace reached the target. After the target maximum temperature was reached, the furnace temperature was maintained for 30 min to achieve better thermal stability of the entire specimen. Then, we turned off the furnace power switch and cooled the sample to room temperature by opening the furnace door before performing the residual strength test. A field emission scanning electron microscope (FE-SEM; JEOL JSM-7401F, Tokyo, Japan) was used to observe the microscopic state of the concrete before and after the high temperature to understand its changes. After the compressive strength test, the fractured cylindrical specimens were retained for SEM observation. An appropriate number of fragments were extracted from the cylindrical specimen and the hydration reaction of the fragments was stopped using the methanol replacement method. These pieces were then placed in an oven at 105 °C and dried for at least 24 h before SEM observation.

## 3. Results and Discussion

### 3.1. Fresh Concrete Properties

The fresh properties of each concrete mixture are listed in Table 7. The table shows that the slump value ranged from 250 to 262 mm. Both the control group and the experimental group had fairly good workability. The HPC-1 mix of the control group had a slump flow of 690 mm, indicating very good fluidity. For the experimental group, the slump flow of each mixture was lower than that of the control group due to the incorporation of fibers, but they also had proper fluidity. Table 7 shows that the unit weight of the control group was 2318 kg/m^3^. For the experimental group, the unit weight ranged from 2274 to 2316 kg/m^3^. Overall, the unit weight ratio between the experimental group and the control group ranged from 0.981 to 0.999, with no significant differences between each other.

### 3.2. Hardened Concrete Properties at Room Temperature

The seven-day results of the hardened properties of the control group and the experimental group, including the compressive strength, elastic modulus, flexural strength, and splitting strength, are shown in Table 8. The data in the table are the average of three specimens.

#### 3.2.1. Compressive Strength of Concrete at Room Temperature

The use of pozzolanic material has a significant impact on the development of UHPC’s early compressive strength, mainly depending on its type, addition amount, mineral composition, particle shape and fineness, and pozzolanic activity. Table 8 shows that the control group had the lowest seven-day compressive strength (81.6 MPa). In the experimental group, the seven-day compressive strength of the UHPC mixtures was between 95.6 and 111.5 MPa. This result showed that the incorporation of pozzolanic materials can enhance the early age compressive strength of UHPC, and the percentage of improvement was between 17.2% and 36.6%. Table 9 shows the percentage of 28-day compressive strength achieved by the UHPC mixture at seven days. For the UHPC-5 mix containing large amounts of silica fume, its seven-day compressive strength was 89% of its 28-day compressive strength. For the HPC-1 mix containing large amounts of cement, its seven-day compressive strength was 91% of 28-day compressive strength.

#### 3.2.2. Elastic Modulus of Concrete at Room Temperature

Figure 2 shows the stress–strain curve of each concrete mixture at seven days of age. Figure 2 shows that the initial slope of the rising branch of the stress–strain curve exhibited a linear relationship before reaching the maximum stress. The elastic modulus of the specimen obtained from the stress–strain curve according to ASTM C469 is listed in Table 10. Table 10 shows that the seven-day elastic modulus of the concrete mixture was between 32.3 and 37.2 GPa. The HPC-1 mixture had the lowest elastic modulus (32.3 GPa) and the UHPC-2 mixture had the highest (37.2 GPa). However, in the experimental group, we found no significant difference in the elastic modulus of the concrete mixtures. We also found that the age of concrete at the time of the test did not significantly affect the elastic modulus. Especially for the UHPC-2 and UHPC-3 mixtures, the elastic modulus at 7 and 28 days were almost equal, as shown in Table 10, because the elastic modulus of concrete during the early stages develops rapidly compared to compressive strength [38]. Myers also pointed out that about 90% or more of the elastic modulus value is achieved within the first 24 h after casting [39].

#### 3.2.3. Flexural Strength of Concrete at Room Temperature

The load versus midspan deflection curves for each group of concrete specimens are shown in Figure 3. The load–displacement curve was linear in the region before the peak point. The control group clearly exhibited brittle failure, as shown in Figure 3. In contrast, all test specimens of the experimental group containing steel fiber showed greater load carrying capacity and better flexing performance than the control group. In other words, the experimental specimens incorporating steel fiber exhibited better toughness because the steel fibers in the experimental group specimens had the ability to bridge cracks, as shown in Figure 4. As a result, the area under the load versus deflection curve up to fracture (i.e., fracture toughness) of the experimental group was significantly better than that of the control group. Table 8 shows that the seven-day flexural strength of the concrete mixtures was between 6.6 and 15.1 MPa. The HPC-1 mix had the lowest flexural strength (6.6 MPa) and the UHPC-2 mix had the highest (15.1 MPa). Overall, the flexural strength of the experimental group was 1.85 to 2.29 times that of the control group. This shows that the steel fiber content is an important factor affecting the flexural strength of concrete.

#### 3.2.4. Splitting Strength of Concrete at Room Temperature

Regarding the splitting strength of concrete, the test results were between 4.4 and 10.8 MPa. The HPC-1 mix had the lowest splitting strength (4.4 MPa) and UHPC-5 mix had the highest (10.8 MPa). Overall, the splitting strength of the experimental group was 1.66 to 2.45 times that of the control group. This shows that the steel fiber content is an important factor affecting the splitting strength of concrete. Figure 5 shows that the control group exhibited brittle failure, and the experimental group maintained its original appearance after reaching the ultimate load without much strain or deformation.

### 3.3. Test Results of Hardened Concrete Properties after Exposure to High Temperatures

To understand the influence of high temperature on the mechanical properties of UHPC, the UHPC-1 specimens of the experimental group were selected for further compressive, flexural, and splitting strength tests after exposure to high temperatures in an electric furnace. Generally, in the fire test, the heating rate of the electric furnace had a strong influence. In this study, we first tried to reach the target temperature (600 °C) with heating rates of 15, 10, 5, and 2 °C/min, and maintained the furnace temperature for an additional hour. However, for all of the above cases, this resulted in specimen spalling, as shown in Figure 6a–d. In particular, the faster the heating rate, the more severe the spalling. Therefore, we changed the heating rate to 2 °C/min to reach the target temperature (500 °C) and held the temperature for 0.5 h, so that the specimen remained intact, as shown in Figure 6e. The maximum fire damage temperature was divided into three types: 300, 400, and 500 °C. After reaching the target temperature, the temperature was held for 0.5 h. Then, the furnace was turned off to allow the specimens to cool naturally to ambient temperature. The age of the UHPC-1 specimen when subjected to high temperature fire damage was 56 days. Table 11 provides the results of the compressive, flexural, and splitting strength tests at this age at room temperature. After heating the UHPC-1 specimen to the target temperature for 0.5 h, it was naturally cooled in an electric furnace and left for 24 h. After that, the post-fire compressive, flexural, and splitting strength tests were performed. The test results are also listed in Table 11.

When the internal temperature of the electric furnace reached the target temperature, the internal temperature of the concrete specimen gradually increased due to heat conduction. In this study, two thermocouples were embedded in a concrete cylindrical specimen with a diameter of 10 cm and a height of 20 cm. One of the thermocouples located at half the height was 3 cm from the surface of the concrete specimen and the other at the core of the specimen was 5 cm from the surface of the concrete specimen to measure the actual temperature inside the specimen. Based on these data, the relationship between the internal temperature of the UHPC-1 specimen and the heating and cooling times of the electric furnace was plotted. After reaching the maximum fire temperature, its holding time was 0.5 h, and then the electric furnace was turned off. When the target temperatures were 300, 400, and 500 °C, the relationship curves between the internal temperature of the concrete specimen and the heating as well as cooling time of the electric furnace were plotted, as shown in Figure 7, Figure 8 and Figure 9. These figures shows that a significant gap between the internal temperature of the specimen and the furnace temperature, which was only about half that of the furnace temperature.

#### 3.3.1. Compressive Strength of Concrete after Exposure to High Temperatures

The results of the 56-day compressive strength of the UHPC-1 specimen are shown in Table 11. At room temperature, the 56-day compressive strength of the UHPC-1 specimen was 155.8 MPa, which is higher than the general requirement for the compressive strength of UHPC to be greater than 150 MPa. Table 11 also lists the residual compressive strength of the UHPC-1 specimen after exposure to different temperatures; the residual compressive strength of the UHPC-1 specimen after exposure to 300, 400, and 500 °C was between 155 and 157 MPa. The relationship between the residual compressive strength and the maximum fire temperature is shown in Figure 10a; the residual compressive strength of the UHPC-1 specimen after exposure to 300, 400, and 500 °C did not decrease significantly, and even increased. The strength of the specimen was slightly increased due to the drying effect of heating [40]. Although the furnace temperature was maintained for 0.5 h, the temperature inside the concrete specimen was still lower than the target temperature. Under the action of high temperature drying, the water vapor in the test specimen was easily released, which helped to improve the compressive strength. Deshpande et al. [41] reported that a slight increase in residual strength compared to room temperature strength may be related to favorable changes in microstructure at high temperatures and inherent variability of concrete materials. At the target temperatures of 300, 400, and 500 °C, Figure 7, Figure 8 and Figure 9 show that the maximum internal temperature of the concrete specimen was approximately 120, 180, and 250 °C, respectively. The temperature difference between 3 and 5 cm from the surface of the concrete specimen was not obvious. Therefore, fire damage had not caused the decay of the concrete matrix. However, when the temperature was 600 °C, the internal temperature of the concrete specimen caused the concrete matrix to decay and spall, so the residual compressive strength dropped sharply to zero. The loss of strength was mainly caused by physical changes at temperatures below 400 °C. However, when the temperature exceeded 600 °C, the decrease in strength was mainly due to chemical degradation in hydrated products and aggregates, which is not directly related to the content and type of fiber [41]. The relative compressive strength ratio is defined as the ratio of the strength after exposure to the target high temperature to the original strength at room temperature. Figure 10b shows that the relative residual compressive strength ratios after exposure to 300 and 400 °C were both greater than one. However, as the temperature reached 500 °C, its relative compressive strength began to decline; especially when the temperature was 600 °C, the relative compressive strength ratio after fire damage was zero.

#### 3.3.2. Flexural Strength of Concrete after Exposure to High Temperatures

At room temperature, the 56-day flexural strength of the UHPC-1 specimen was 13 MPa, which was significantly higher than that of ordinary concrete (Table 11). Table 11 also lists the residual flexural strength of the UHPC-1 specimen after exposure to different temperatures. The residual flexural strength of the UHPC-1 specimen after exposure to 300, 400, and 500 °C was between 10.3 and 13.3 MPa. The relationship between the residual flexural strength and the fire temperature is depicted in Figure 11a; the residual flexural strength of the UHPC-1 specimen after exposure to 300, 400, and 500 °C did not decrease significantly, and even increased. The strength of the specimen slightly increased due to the drying effect of heating. However, when the target temperature was 600 °C, spalling occurred, so the residual flexural strength suddenly reduced to zero. Figure 11b also shows that the relative residual flexural strength ratios after exposure to 300 and 400 °C were all greater than one. However, as the target temperature was 500 °C, its residual flexural strength was significantly reduced, and its relative flexural strength ratio after fire damage is 0.79; when the target temperature was 600 °C, its relative flexural strength was zero.

#### 3.3.3. Splitting Strength of Concrete after Exposure to High Temperatures

The results of the 56-day splitting strength of the UHPC-1 specimen are shown in Table 11. At room temperature, the 56-day splitting strength of the UHPC-1 specimen was 9.8 MPa, which is significantly higher than that of ordinary concrete. Table 11 also lists the residual splitting strength of the UHPC-1 specimen after exposure to different temperatures. The residual splitting strength of the UHPC-1 specimen after exposure to 300, 400, and 500 °C was between 9.2 and 9.9 MPa, and the relationship between the residual splitting strength and the target temperature is shown in Figure 12a; the residual cleavage strength of the UHPC-1 specimen after 300, 400, and 500 °C did not decrease significantly, and even increased. However, when the temperature was 600 °C, spalling occurred, so the residual splitting strength was zero. Figure 12b shows that the relative splitting strength ratio after exposure to 300 °C was greater than one. However, as the target temperature reached 400 and 500 °C, its residual splitting strength showed a decline. The relative splitting strength ratio after fire damage was 0.94. When the target temperature was 600 °C, the relative splitting strength ratio after fire damage quickly dropped to zero.

#### 3.3.4. SEM Observation before and after Exposure to High Temperatures

Figure 13 compares the morphological characteristics of the UHPC-1 specimen before and after exposure to high temperatures. The micrographs captured from the core of the cylindrical specimens revealed hydrated phases such as C–S–H gels. The addition of silica fume and ultra-fine silica powder resulted in a much denser microstructure due to the pozzolanic reaction between supplementary cementitious materials and cement hydration product (calcium hydroxide) (Figure 13). In Figure 13a, the magnification is 50×, and the well-adhered aggregates in the UHPC matrix were uniformly distributed. In addition, at room temperature, an excellent bond between the cementitious material and the fine aggregate were observed in the test piece, and the formation of ettringite (Al_2_O_3_–Fe_2_O_3_–tri) and silicate was detected in the pores present. After exposure to high temperatures, the microstructure and properties of the UHPC matrix changed with increasing temperature, as shown in Figure 13b. Figure 13c shows that as the temperature increased, the bonding interface between the UHPC matrix and the steel fibers gradually loosened. In particular, when the target temperature was 500 °C, the steel fibers were pulled out or broken after the peak loading Figure 14 shows that although the target temperature had been raised to 500 °C, typical acicular AFt crystals were still observed that withstood the thermal aggression in a high-temperature environment.

In general, when the temperature inside concrete reaches 105–200 °C, C–S–H gels begin to lose bound water and undergo chemical changes [19,20]. When the target temperature was 300–400 °C, the maximum internal temperature of the concrete specimen was approximately 180–200 °C, as shown in Figure 7 and Figure 8. The test pieces in Figure 13 show that due to heating, a part of ettringite and C–S–H gels were destroyed and dispersed. In addition, the formation of acicular ettringite was round and cracks occurred due to thermal expansion and contraction. Notably, the cracks in the UHPC matrix after exposure to 300 and 400 °C, as shown in Figure 13, were narrower than those of the matrix at room temperature. The reduction of crack width in the matrix after exposure to 300 C and 400 °C is attributed to the drying effect of heating that occurred under exposure temperatures. Therefore, the increase in compressive strength of the UHPC-1 specimen can be improved up to 400 °C. However, once the internal temperature of the concrete was above 200 °C, the decomposition of cement hydrate and the decay of aggregate gradually occurred [19,20]. SEM observations of the UHPC-1 specimen exposed to 500 °C exhibited that the microstructure of concrete had changed due to microcracks, voids, partial C–S–H gels deterioration, and calcium hydroxide crystal deformation. When the target temperature was 500 °C, the maximum internal temperature of the concrete specimen was approximately 250 °C, as shown in Figure 9. At this temperature, most of the internal bound water of hydrates containing Al_2_O_3_ or Fe_2_O_3_ was lost; about 20% of the bound water of C–S–H gels was also lost [19,20]. At the same time, the polypropylene fibers in the test piece melted. Therefore, the holes left after the polypropylene fibers melted can be observed in Figure 13. The size of these holes was consistent with the size of the polypropylene fibers and had a certain depth. This result shows that polypropylene fibers melt at high temperatures and a network of pores is created, helping to reduce internal vapor pressure and maintaining a certain residual strength. In other words, the SEM observations confirmed that the melting of the polypropylene fibers resulted in an increase in the permeability of the concrete matrix, thereby releasing water vapor. This result is consistent with the findings of Dong et al. [23], Hager et al. [24], and Wu et al. [25]. However, as the temperature increased further, cement hydrate decomposition and aggregate decay was more serious. The difference in the thermal deformation between the UHPC matrix and the aggregate caused stress concentration, so the strength of the concrete began to decrease. As mentioned above, when the target temperature was 600 °C, the number of cracks increased and the cracks widened, which weakened the internal structure of the UHPC-1 specimen and eventually reduced the compressive strength. From the above, SEM investigations produced evidence of the various types of damage caused by exposure to fire.

## 4. Conclusions

In this study, we explored the mechanical properties of UHPC before and after exposure to high temperatures. Based on the analysis and discussion of the test results, the results obtained are summarized as follows. At room temperature, the 56-day compressive strength of the UHPC-1 mix was 155.8 MPa, which is higher than the general compressive strength requirement of 150 MPa for UHPC. After the UHPC-1 specimen was exposed to 300, 400, and 500 °C, the residual compressive strength did not decrease significantly, and even increased. The strength of the specimen increased slightly due to the drying effect of temperature rise. When the target temperatures were 300, 400, and 500 °C, the maximum internal temperature of the concrete specimen was about 120, 180, and 250 °C, respectively. Therefore, the fire damage did not cause the decay of the concrete matrix. However, when the fire damage was 600 °C, the internal temperature of the concrete specimen had caused the concrete matrix to decay and spall. In terms of flexural strength, the 56-day strength of the UHPC-1 mix at room temperature was 13 MPa, which was significantly higher than that of ordinary concrete. After the UHPC-1 specimen was exposed to 300, 400, and 500 °C, the residual flexural strength did not decrease significantly, and even increased slightly due to the drying effect of heating. The relative residual flexural strength ratios after fire damage at 300 and 400 °C were both greater than one. However, as the target temperature of the fire damage reached 500 °C, the residual flexural strength of the UHPC-1 specimen decreased significantly, and its residual flexural strength ratio after fire damage was 0.79. For the splitting strength, the 56-day splitting strength of the UHPC-1 mix at normal temperature was 9.8 MPa, which is also significantly higher than that of ordinary concrete. After the UHPC-1 specimen was exposed to 300, 400, and 500 °C, the residual splitting strength did not decline significantly, and even increased slightly due to the drying effect of heating. The relative residual splitting strength ratio after fire damage at 300 °C was greater than one. However, as the fire damage temperature reached 400 and 500 °C, the residual splitting strength declined, and the residual splitting strength ratio after fire damage was 0.94. SEM observations confirmed that polypropylene fibers melted at high temperatures, forming other channels that helped to reduce the internal vapor pressure of UHPC and maintain a certain residual strength. However, the optimal amount of polypropylene fiber needs to be further explored.

## Figures and Tables

**Figure 1 materials-13-00770-f001:**
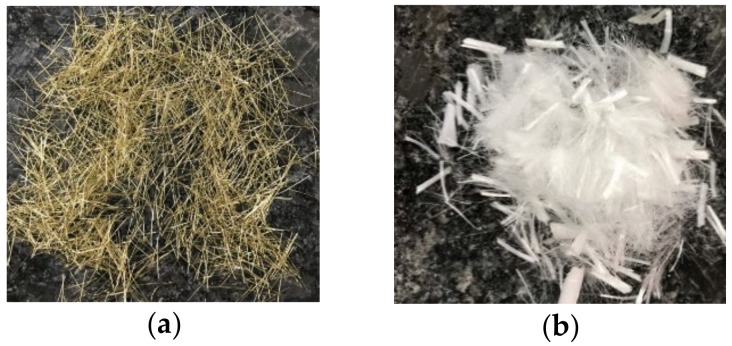
Appearance of (**a**) steel fibers and (**b**) polypropylene fibers.

**Figure 2 materials-13-00770-f002:**
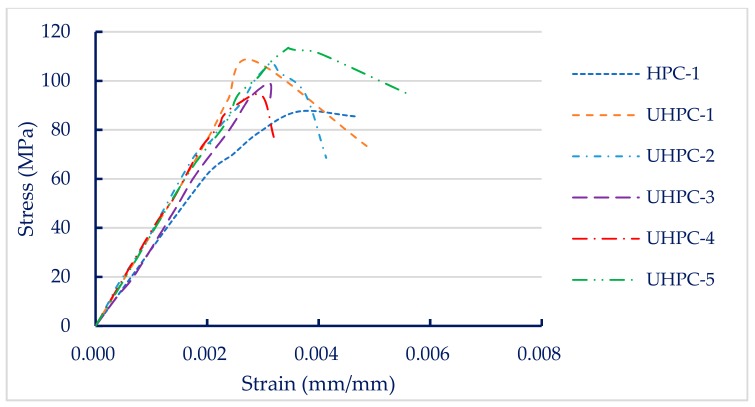
Stress–strain curves of concrete.

**Figure 3 materials-13-00770-f003:**
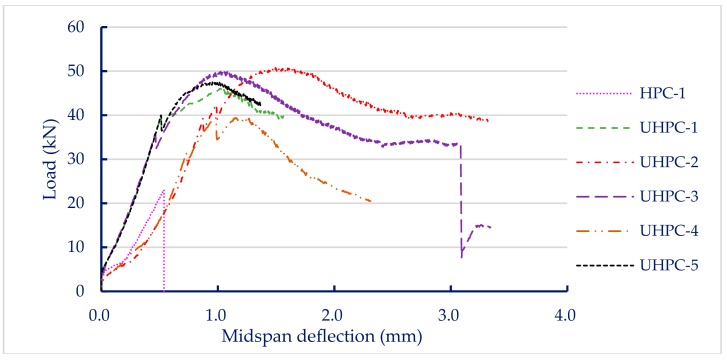
Load–deflection curves of concrete specimens.

**Figure 4 materials-13-00770-f004:**
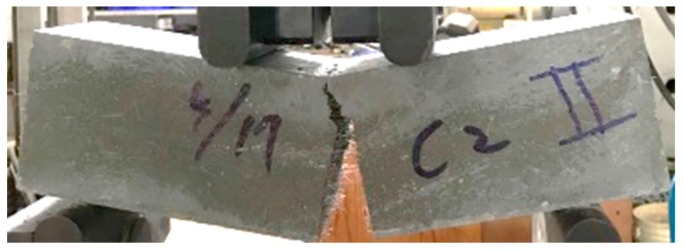
Failure of the experimental group specimen in flexure test.

**Figure 5 materials-13-00770-f005:**
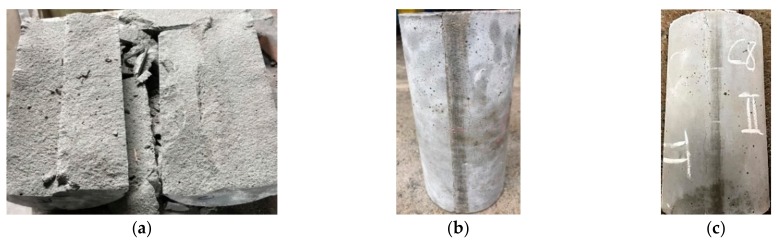
Comparison of splitting test results: (**a**) HPC-1, (**b**) UHPC-3, and (**c**) UHPC-5 specimens.

**Figure 6 materials-13-00770-f006:**
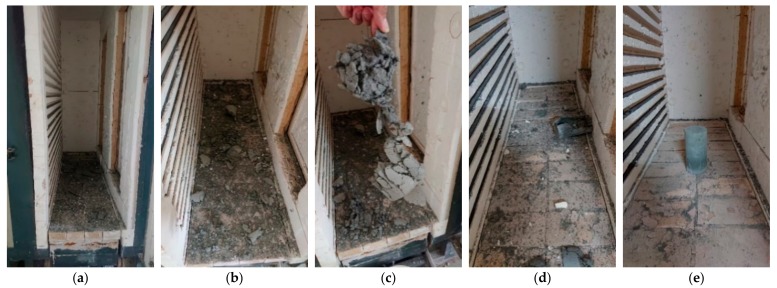
Rupture of the specimen after high temperature: (**a**) 15 °C/min heating rate and target temperature of 600 °C, (**b**) 10 °C/min heating rate and target temperature of 600 °C, (**c**) 5 °C/min heating rate and target temperature of 600 °C, (**d**) 2 °C/min heating rate and target temperature of 600 °C, and (**e**) 2 °C/min heating rate and target temperature of 500 °C.

**Figure 7 materials-13-00770-f007:**
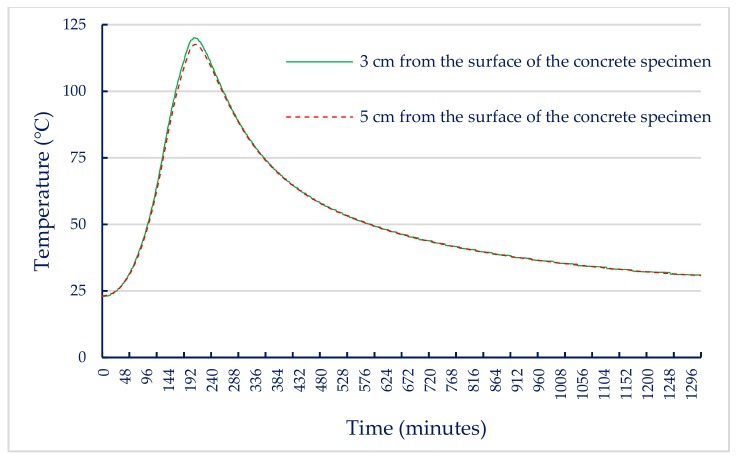
Relationship between specimen internal temperature and time (target temperature = 300 °C).

**Figure 8 materials-13-00770-f008:**
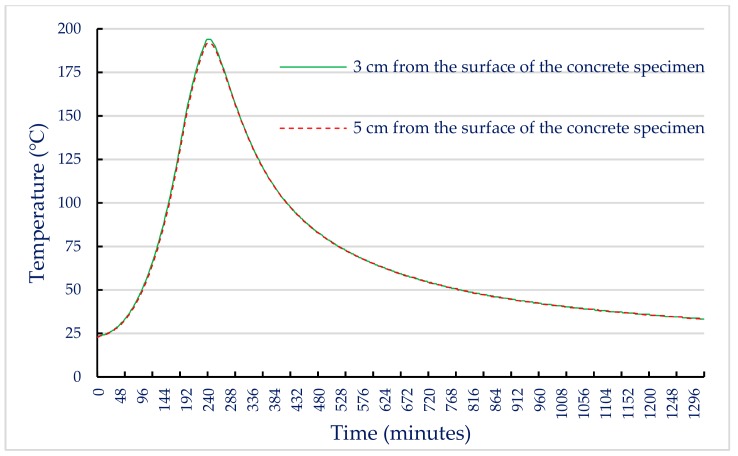
Relationship between specimen internal temperature and time (target temperature = 400 °C).

**Figure 9 materials-13-00770-f009:**
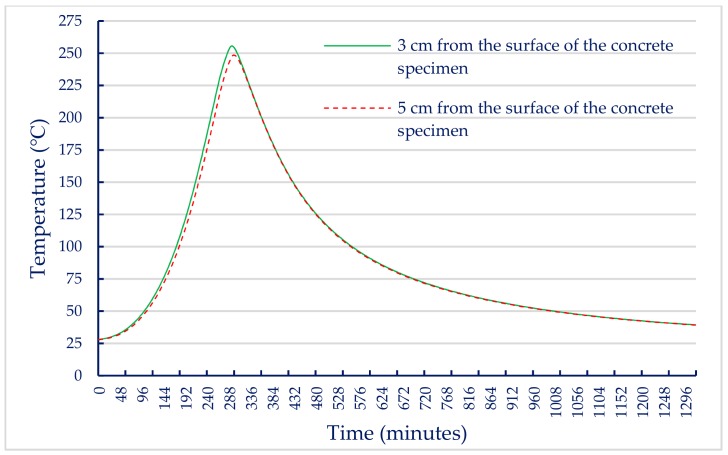
Relationship between specimen internal temperature and time (target temperature = 500 °C).

**Figure 10 materials-13-00770-f010:**
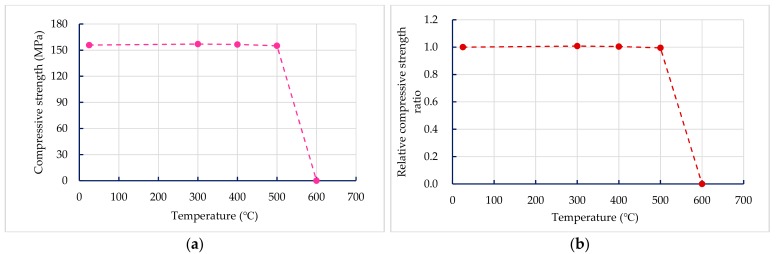
Comparison of compressive strength before and after fire damage to the UHPC-1 specimen: (**a**) residual compressive strength after high temperature; (**b**) relative compressive strength ratio.

**Figure 11 materials-13-00770-f011:**
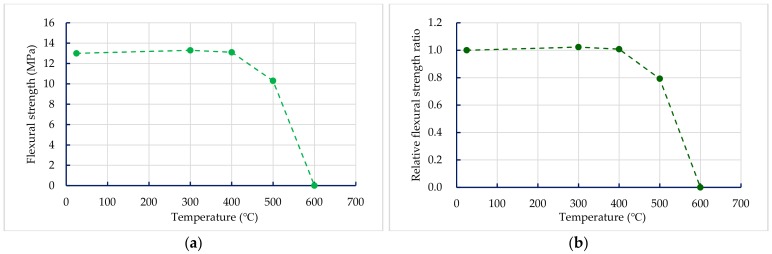
Comparison of flexural strength before and after fire damage to the UHPC-1 specimen: (**a**) residual flexural strength after high temperature; (**b**) relative flexural strength ratio.

**Figure 12 materials-13-00770-f012:**
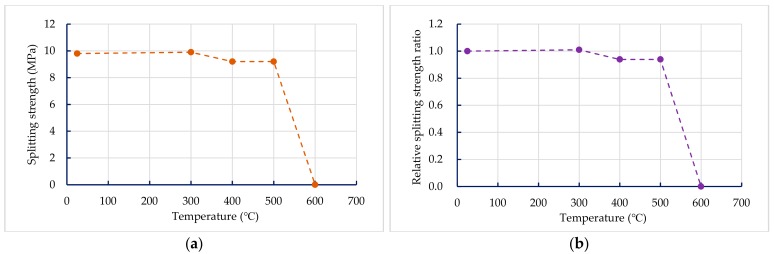
Comparison of splitting strength before and after fire damage to the UHPC-1 specimen: (**a**) residual splitting strength after high temperature; (**b**) relative splitting strength ratio.

**Figure 13 materials-13-00770-f013:**
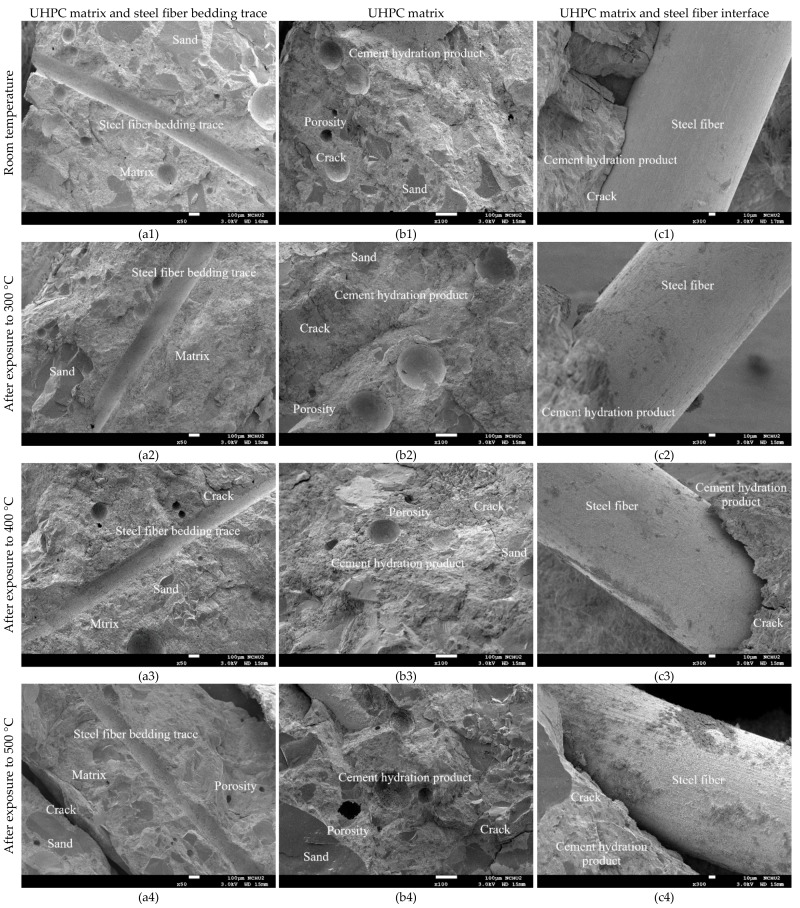
Comparison of SEM observation before and after fire damage: (**a**) 50× (**b**) 100×, and (**c**) 300× magnification.

**Figure 14 materials-13-00770-f014:**
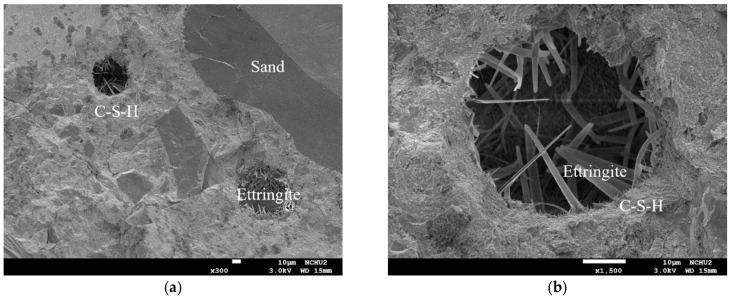
SEM observation after exposure to 500 °C: (**a**) 300× and (**b**) 1500× magnification.

**Table 1 materials-13-00770-t001:** Physical properties and chemical composition of fine aggregates.

Type of Quartz Sand	Physical Properties		Chemical Composition		
	Specific Gravity (S.S.D.)	Water Absorption Rate (%) (S.S.D.)	SiO_2_ (%)	Fe_2_O_3_ (%)	Al_2_O_3_ (%)
Type I	2.65	≈0	99.82	0.014	0.033
Type II	2.65	≈0	99.84	0.016	0.034

Notes: S.S.D., Saturated-Surface-Dry.

**Table 2 materials-13-00770-t002:** Particle size distribution of fine aggregates.

Sieve No. (ASTM E11-70)	Particle Size (μm)	Percentage Retained (%)	
		Type I	Type II
20	850	0.04	-
30	600	20.15	-
40	425	67.83	-
50	300	11.81	-
60	250	-	0.05
70	212	0.17	13.69
100	150	-	36.55
140	106	-	32.39
200	75	-	12.73
270	53	-	3.61

**Table 3 materials-13-00770-t003:** Basic properties of fibers.

Type of Fiber	Length (mm)	Diameter (mm)	Density (g/cm^3^)	Elastic Modulus (GPa)	Tensile Strength (MPa)	Melting Point (°C)
Steel Fibers	13	0.2	7.8	200	2000	-
Polypropylene Fibers	12	0.05	0.9	-	300	165

**Table 4 materials-13-00770-t004:** Mix proportions of concrete.

Mix No.	W/B	W (kg/m^3^)	C (kg/m^3^)	SF (kg/m^3^)	SFP (kg/m^3^)	SP (kg/m^3^)	VA (kg/m^3^)	PP (kg/m^3^)	Steel Fiber (kg/m^3^)	FA (kg/m^3^)
HPC-1	0.195	196	1005	0	0	26	1	0	0	1286
UHPC-1	0.195	187	774	167	21	25	1	0.3	39	1231
UHPC-2	0.195	186	756	179	21	25	1	0.5	78	1223
UHPC-3	0.195	186	747	179	31	25	1	0.3	59	1225
UHPC-4	0.195	186	744	191	21	25	1	0.8	59	1223
UHPC-5	0.195	186	737	191	26	25	1	0.3	78	1220

Notes: HPC, high performance concrete; UHPC, ultra-high performance concrete; W/B, water-binder ratio; W, water; C, cement; SF, silica fume; SFP, ultra-fine silica powder; S, superplasticizers; VA, viscous agent; PP, polypropylene fiber; FA, fine aggregate.

**Table 5 materials-13-00770-t005:** UHPC test parameters.

Mix No.	Parameter			
	Percentage of Cement Replaced by SF (%)	Percentage of Cement Replaced by SFP (%)	Amount of PP (Volume Percent) (%)	Amount of Steel Fiber (Volume Percentage) (%)
UHPC-1	17.4	2.2	0.03	0.5
UHPC-2	18.7	2.2	0.06	1.0
UHPC-3	18.7	3.2	0.03	0.75
UHPC-4	20.0	2.2	0.09	0.75
UHPC-5	20.0	2.7	0.03	1.0

**Table 6 materials-13-00770-t006:** Concrete property test method.

Property	Experimental Method
Unit weight	ASTM C138
Slump	ASTM C143
Slump flow	ASTM C1611
Compressive strength	ASTM C39
Elastic modulus	ASTM C469
Splitting tensile strength	ASTM C496
Flexural strength	ASTM C78

**Table 7 materials-13-00770-t007:** Results of fresh concrete properties.

Mix No.	Slump (mm)	Slump Flow (mm)	Unit Weight (kg/m^3^)
HPC-1	262	690	2318
UHPC-1	255	530	2263
UHPC-2	257	540	2316
UHPC-3	260	490	2274
UHPC-4	255	530	2288
UHPC-5	250	510	2308

**Table 8 materials-13-00770-t008:** Results of hardened concrete properties at room temperature.

Mix No.	7-day Experimental Results			
	*f_c_*′ (MPa)	*E_c_* (GPa)	*f_r_* (MPa)	*f_s_* (MPa)
HPC-1	81.6 (7.8)	32.3	6.6 (0.5)	4.4 (0.2)
UHPC-1	95.7 (11.2)	34.9	12.6 (1.4)	7.3 (0.9)
UHPC-2	107.0 (2.8)	37.2	15.1 (0.2)	9.6 (0.8)
UHPC-3	97.8 (7.9)	33.9	14.7 (0.3)	10.5 (0.3)
UHPC-4	95.6 (1.5)	35.4	12.2 (0.6)	8.9 (0.4)
UHPC-5	111.5 (3.3)	36.9	12.7 (1.4)	10.8 (0.1)

Note: *f_c_*’, compressive strength; *E_c_*, elastic modulus; *f_r_*, flexural strength; *f_s_*, splitting strength; data in parentheses are standard deviation.

**Table 9 materials-13-00770-t009:** Development of compressive strength.

Mix No.	$f_{c-7d}^{\prime }(\mathrm{MPa})$	$f_{c-28d}^{\prime }(\mathrm{MPa})$	$f_{c-7d}^{\prime }$ $\;/f_{c-28d}^{\prime }$
HPC-1	81.6 (7.8)	90.0 (3.4)	0.91
UHPC-1	95.7 (11.2)	128.1 (5.4)	0.75
UHPC-2	107.0 (2.8)	127.0 (5.4)	0.84
UHPC-3	97.8 (7.9)	121.6 (14.9)	0.80
UHPC-4	95.6 (1.5)	123.6 (4.4)	0.77
UHPC-5	111.5 (3.3)	125.8 (6.8)	0.89

Note: $f_{c-7d}^{\prime }$, 7-day compressive strength; $f_{c-28d}^{\prime }$, 28-day compressive strength; data in parentheses are standard deviation.

**Table 10 materials-13-00770-t010:** Development of elastic modulus.

Mix No.	*Ec_-7D_* (GPa)	*Ec_-28D_* (GPa)	*Ec_-7D_*/*Ec_-28D_*
HPC-1	32.3	37.2	0.87
UHPC-1	34.9	39.5	0.88
UHPC-2	37.2	38.1	0.98
UHPC-3	33.9	34.0	1.00
UHPC-4	35.4	48.0	0.74
UHPC-5	36.9	44.8	0.82

Note: *E_c-7D_*, 7-day elastic modulus; *E_c-28D_*, 28-day elastic modulus.

**Table 11 materials-13-00770-t011:** Results of 56-day hardened concrete properties at room temperature.

Target Temperature	*f_c_*′ (MPa)	*f_r_* (MPa)	*f_s_* (MPa)
Room	155.8 (8.8)	13.0 (0.28)	9.8 (0.39)
300 °C	157.0 (7.3)	13.3 (1.33)	9.9 (0.98)
400 °C	156.5 (9.7)	13.1 (1.07)	9.2 (0.85)
500 °C	155.0 (8.9)	10.3 (1.24)	9.2 (0.27)

Note: *f_c_*’, compressive strength; *f_r_*, flexural strength; *f_s_*, splitting strength; data in parentheses are SD.

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
