# Peer review of "Mechanical Properties of Ultra-High Performance Concrete before and after Exposure to High Temperatures"

_materials, 2020, doi:10.3390/ma13030770_

Round 1
Reviewer 1 Report
This research introduced the changes in mechanical propertes of UHPC before and after expose to high temperature. The article was well written and the interest results were shown. It can be more informable by adding more results.
The authors choose the UHPC(1 to 5) with PP and Steel fiber. Please explain why the authors used these mix proportions. The authors explain the reason of decline of mechanical properties after expose to high temperature were caused by the melting of PP fibers. It can be more reliable if this paper could show the comparison result between the UHPC with PP fibers and without PP fibers(only steel fibers). Actually, the melting point of PP fibers are 165 degree Celcius. The maximum temperature of 3cm from the concrete specimen was higer than 165 ℃ at targer temperature was over 400℃. Thus, it will be more helpful to explain the reason of degradation of concrete if the author can provide the variations in internal temperature of concrete specimens.
Author Response
Response to Reviewer 1 Comments
This research introduced the changes in mechanical propertes of UHPC before and after expose to high temperature. The article was well written and the interest results were shown. It can be more informable by adding more results.
Point 1: The authors choose the UHPC (1 to 5) with PP and Steel fiber. Please explain why the authors used these mix proportions.
Response: This study screened out four important parameters, including the percentage of cement replaced by silica fume (SF), the percentage of cement replaced by ultra-fine silica powder (SFP), the amount of steel fiber (volume percent), and the amount of polypropylene fiber (volume percentage). Table 5 lists the percentage of these parameters used in the mix proportions of the UHPC. In other words, the five different UHPC mix proportions in the experimental group have different amounts of steel fiber and polypropylene fiber. In addition, the amount of superplasticizers and viscous agents are 2.6% and 0.1% by weight of the cementitious materials, respectively. The changes in the test parameters listed in Table 5 were used to explore their effect on the concrete properties in the experimental group.
Point 2: The authors explain the reason of decline of mechanical properties after expose to high temperature were caused by the melting of PP fibers. It can be more reliable if this paper could show the comparison result between the UHPC with PP fibers and without PP fibers (only steel fibers).
Response: The authors thank the reviewer for his valuable comments. The comparison results of UHPC with and without PP fiber (steel fiber only) are planned to be presented in another article. Therefore, this article only discusses the results of UHPC with PP fibers and steel fibers.
Point 3: Actually, the melting point of PP fibers are 165 degree Celcius. The maximum temperature of 3 cm from the concrete specimen was higer than 165 ℃ at target temperature was over 400 ℃. Thus, it will be more helpful to explain the reason of degradation of concrete if the author can provide the variations in internal temperature of concrete specimens.
Response: The authors thank the reviewer for his valuable comments. In this study, only two thermocouples were embedded in a concrete cylindrical specimen with a diameter of 10 cm and a height of 20 cm. One of the thermocouples located at half the height is 3 cm from the surface of the concrete specimen and the other at the core of the specimen is 5 cm from the surface of the concrete specimen in order to measure the actual temperature inside the specimen. However, it can be known from these thermocouples that the internal temperature of the specimen was far from the furnace temperature, which was only about half of the furnace temperature. In order to know more precisely the variations in the internal temperature of concrete specimens, more thermocouples will be buried in subsequent research, which will help explain the cause of concrete degradation.
Point 4: English language and style are fine/minor spell check required
Response: The English writing of the revised manuscript has been reviewed by professionals.

Reviewer 2 Report
The article is devoted to the recognition of the behavior of ultra high-quality concrete (UHPC) in high temperature conditions. Therefore, the authors examined the properties of UHPC with a designed 28-day compressive strength of 120 MPa or more in the fresh mix phase, and its hardened mechanical properties were measured at the age of 7 days. Test variables included: type of cement and mixing ratio (silica ash, ultra-fine silicon powder), type of fiber (steel fiber, polypropylene fiber) and fiber content (volume percent).
The substantive layout of the work and the research program can be assessed as very good. I have noticed some glitches that I suggest authors consider:
1. Chapter 1: There is a lack of UHPC applications - urban, industrial or bridges - exposed to fire. Whether in real constructions there was a failure of reinforced concrete structures caused by high temperature. If so, which ones? Absolutely please emphasize the novelty in the research carried out and the contribution to the development of concrete technology.
2. Chapter 3.2.1. In table 9 please provide the values ​​of standard deviations of the results.
Chapter 3.3. Why was the high temperature maintained for 0.5 hours? Is this due to fire protection requirements?
Chapter 3.3.1. Fig. 10-12 shows changes in strength as a function of temperature. I think that linear approximation of end parts is not allowed. It's best to mark this part with a dotted line.
Chapter 3.3.4. If it is stated that the C-S-H gel and Al2O3 or Fe2O3 compounds have deteriorated, please indicate on microscopic photographs exactly what these compounds looked like before and after the temperature treatment.
Chapter 4. Can you make a specific recommendation regarding the need to protect concrete with PP fibers, e.g. REI 120, REI 240 (according to EN 1996-1-2: 2010 or according to the corresponding US standard).
Author Response
Response to Reviewer 2 Comments
The article is devoted to the recognition of the behavior of ultra high-quality concrete (UHPC) in high temperature conditions. Therefore, the authors examined the properties of UHPC with a designed 28-day compressive strength of 120 MPa or more in the fresh mix phase, and its hardened mechanical properties were measured at the age of 7 days. Test variables included: type of cement and mixing ratio (silica ash, ultra-fine silicon powder), type of fiber (steel fiber, polypropylene fiber) and fiber content (volume percent).
The substantive layout of the work and the research program can be assessed as very good. I have noticed some glitches that I suggest authors consider:
Point 1: Chapter 1: There is a lack of UHPC applications - urban, industrial or bridges - exposed to fire. Whether in real constructions there was a failure of reinforced concrete structures caused by high temperature. If so, which ones? Absolutely please emphasize the novelty in the research carried out and the contribution to the development of concrete technology.
Response: The use of UHPC in infrastructure projects has increased in recent years. However, previous research studies have shown that UHPC may not exhibit good fire performance and is susceptible to fire-induced spalling. The behavior of concrete during a fire mainly depends on its mix proportions and constituents and is determined by complex physicochemical transformations occurring in heating. When exposed to high temperatures, ordinary strength concrete and high-performance concrete follow similar trends in microstructure, but UHPC behaves differently. Essentially, UHPC is produced primarily through the use of a relatively low water/cementitious ratio and incorporates silica fume. This leads to a reduced permeability relative to ordinary concretes. Therefore, when exposed to high temperatures, UHPC is prone to explosive spalling due to its increased packing density. In view of this, this article explores the properties of UHPC before and after high temperatures. Based on the reviewer's recommendations, the novelty of the research undertaken and its contribution to the development of concrete technology have been emphasized in the first chapter of the revised manuscript.
Point 2: Chapter 3.2.1. In table 9 please provide the values of standard deviations of the results.
Response: According to the reviewer's suggestion, the standard deviation values of the test results have been listed in the revised manuscript.
Point 3: Chapter 3.3. Why was the high temperature maintained for 0.5 hours? Is this due to fire protection requirements?
Response: After reaching the target maximum temperature, the furnace temperature was maintained for 30 minutes. The main purpose was to make the temperature inside the entire specimen closer to the target temperature.
Point 4: Chapter 3.3.1. Fig. 10-12 shows changes in strength as a function of temperature. I think that linear approximation of end parts is not allowed. It's best to mark this part with a dotted line.
Response: In Figures 10-12 of the revised manuscript, dotted lines have been used instead.
Point 5: Chapter 3.3.4. If it is stated that the C-S-H gel and Al2O3 or Fe2O3 compounds have deteriorated, please indicate on microscopic photographs exactly what these compounds looked like before and after the temperature treatment.
Response: According to the reviewer's suggestion, the appearance of these compounds before and after high temperature has been marked on the SEM photos in the revised manuscript.
Point 6: Chapter 4. Can you make a specific recommendation regarding the need to protect concrete with PP fibers, e.g. REI 120, REI 240 (according to EN 1996-1-2: 2010 or according to the corresponding US standard).
Response: Failure of concrete structural elements at high temperatures may be due to loss of bending or tensile strength; loss of bonding strength; loss of shear or torsional strength; loss of compressive strength; and spalling of concrete. Therefore, a structural element should be designed to fulfill its separating and/or load-bearing function without failure for the required period of time in a given fire scenario. Design for fire resistance aims to ensure overall dimensions of the section of an element sufficient to keep the heat transfer through this element within acceptable limits and an average concrete cover to the reinforcement sufficient to keep the temperature of the reinforcement below critical values long enough for the required fire resistance period to be attained. According to EN 1996-1-2, the fire-resistance rating of a structure is identified by the REI marking, which is composed of the following elements: R=load-bearing, E=integrity, and I=thermal insulation. Specific proposals for the use of PP fibers to protect UHPC must be completed through a complete plan and then through fire resistance tests. Therefore, the experimental results of this study are not enough to make specific recommendations. However, the test results have confirmed that PP fibers can prevent UHPC from explosive spalling at high temperatures. According to the reviewer's suggestion, the conclusions have been appropriately modified in the revised manuscript.
